# Temporal Artery Vascular Diseases

**DOI:** 10.3390/jcm11010275

**Published:** 2022-01-05

**Authors:** Hélène Greigert, André Ramon, Georges Tarris, Laurent Martin, Bernard Bonnotte, Maxime Samson

**Affiliations:** 1Department of Internal Medicine and Clinical Immunology, Dijon University Hospital, 21000 Dijon, France; helene.greigert@chu-dijon.fr (H.G.); bernard.bonnotte@chu-dijon.fr (B.B.); 2Department of Vascular Medicine, Dijon University Hospital, 21000 Dijon, France; 3Interactions Hôte-Greffon-Tumeur/Ingénierie Cellulaire et Génique, LabEx LipSTIC, INSERM, EFS BFC, UMR1098, Université de Bourgogne Franche-Comté, 21000 Dijon, France; 4Department of Rheumatology, Dijon University Hospital, 21000 Dijon, France; andre.ramon@chu-dijpon.fr; 5Department of Pathology, Dijon University Hospital, 21000 Dijon, France; georges.tarris@chu-dijon.fr (G.T.); laurent.martin@chu-dijon.fr (L.M.)

**Keywords:** temporal arteritis, giant cell arteritis, ANCA-associated vasculitis, varicella-zoster virus

## Abstract

In the presence of temporal arteritis, clinicians often refer to the diagnosis of giant cell arteritis (GCA). However, differential diagnoses should also be evoked because other types of vascular diseases, vasculitis or not, may affect the temporal artery. Among vasculitis, Anti-neutrophil cytoplasmic antibodies (ANCA)-associated vasculitis is probably the most common, and typically affects the peri-adventitial small vessel of the temporal artery and sometimes mimics giant cell arteritis, however, other symptoms are frequently associated and more specific of ANCA-associated vasculitis prompt a search for ANCA. The Immunoglobulin G4-related disease (IgG4-RD) can cause temporal arteritis as well. Some infections can also affect the temporal artery, primarily an infection caused by the varicella-zoster virus (VZV), which has an arterial tropism that may play a role in triggering giant cell arteritis. Drugs, mainly checkpoint inhibitors that are used to treat cancer, can also trigger giant cell arteritis. Furthermore, the temporal artery can be affected by diseases other than vasculitis such as atherosclerosis, calcyphilaxis, aneurysm, or arteriovenous fistula. In this review, these different diseases affecting the temporal artery are described.

## 1. Introduction

The term “temporal arteritis” is sometimes used to refer to giant cell arteritis (GCA) but this term is not appropriate. In fact, GCA does not consistently affect the temporal artery (TA) and other types of vasculitis or non-inflammatory diseases may affect TA. In this review, all these different diseases affecting the temporal artery are described.

## 2. Temporal Arteritis 

### 2.1. Giant Cell Arteritis (GCA) 

GCA remains of course the first cause of temporal arteritis. GCA is a granulomatous large vessel vasculitis involving large vessels, mainly the aorta and extracranial branches of the external carotid artery such as the TA [1]. GCA is the most common vasculitis in adults. It affects patients after 50 years, with a peak incidence between 70 and 80 years [2]. 

Two types of symptoms are distinguished in GCA (Figure 1). On one hand, systemic signs are frequent, unspecific, and related to systemic inflammation, and correlated to interleukin-6 (IL-6) production. Fever is usually low-grade and reaches 39 °C in only 15% of cases, usually without shearlings. It might be the only feature of GCA. Malaise, anorexia, and weight loss are also common. Around 40% of patients also experience polymyalgia rheumatica (PMR), which is characterized by aching and stiffness of the neck, shoulder, and pelvic girdles. The pain is bilateral and symmetrical, and usually radiates towards the elbows and knees, and predominates in the morning. On the other hand, patients have ischemic signs which are more specific to GCA and can cause morbidity and mortality. The type of ischemic signs directly depends on the topography of the arterial involvement and is the consequence of vascular remodeling that leads to thickening of the vascular wall and ultimately to stenosis or occlusion of the affected arteries [3]. Among them, a new-onset headache is the most frequent symptom, occurring in at least two-thirds of patients. The pain usually occurs over the temporal or occipital areas or diffuse and is poorly released by paracetamol. It can be so intense that it can cause insomnia. Widespread headache should be considered as the first symptom of GCA in patients with PMR [4].

Other cephalic ischemic signs include jaw claudication, scalp tenderness, and, more rarely, scalp or tongue necrosis (Figure 1B). Transient or permanent vision loss may also occur. In most cases, it occurs in patients at diagnosis, exceptionally during a relapse, unless the disease has been poorly treated. The most frequent visual impairment is acute anterior ischemic optic neuropathy and more rarely the occlusion of the central retinal artery or posterior ischemic optic neuropathy. These conditions are the most severe form of GCA, as they constantly leave heavy visual sequelae [5]. Some patients may also present diplopia, transient or permanent, which is often linked to an attack on the 3rd cranial pair [6]. In this case, patients often present with ptosis. Unlike other visual impairments, diplopia usually recovers after a few weeks of treatment with glucocorticoids. In 7% of cases, patients also present with a stroke that preferentially affects men and most often concerns the vertebrobasilar territory [7,8,9].

Two main phenotypes of GCA are usually distinguished and can be mixed together. Cephalic GCA was first described by Bayard Horton in 1932 [10]. It is the typical form of GCA that mainly affects extracranial branches of the carotid artery among which is the TA which is associated with the highest risk of vascular involvement [11]. Clinical examination of the TA typically shows an induration of the TA, which may be tender to palpation with occasional facing edema (Figure 1A). The specificity for the diagnosis of GCA is then greater than 99% [2]. The temporal pulses may also be decreased or even abolished, but the diagnostic specificity of this clinical sign is lower [2].

In addition, recent advances in vascular imaging (angio-computed tomography [CT] scan, positron emission tomography [PET]-CT, angio-MRI) have demonstrated that 30–70% of GCA patients also have involvement of extracranial large vessels, in particular, the aorta and the subclavian or axillary arteries [12]. Its prevalence in GCA has increased with the increasing use of imaging [13,14,15] showing a halo sign by Doppler ultrasound, arterial wall thickening sometimes with contrast enhancement with a CT scan or an angio-MRI, and arterial wall uptake of ^18^Fluoro deoxyglucose on a PET-CT (Figure 1C,D) [13,14,15,16,17]. 

Color Doppler ultrasound of the TA and supra-aortic arteries is the first-line examination in case of suspected GCA because it is less invasive, less costly, and has a lower rate of false negatives than a temporal artery biopsy (TAB) [18]. The halo sign, which is investigated by color Doppler ultrasound and is defined as a homogeneous, hypoechoic thickening of the arterial wall, visible in both the longitudinal and transverse planes and not compressible, has a sensitivity and specificity of 68% and 81%, respectively (Figure 2) [19]. Thus, there are halo signs without GCA (false positives), with an estimated rate of 4.6% in the study by Fernandez-Fernandez [20]. Among 14 patients with a halo sign without GCA, 29% had PMR, 21% had atherosclerosis, and the remaining 50% had a wide variety of diagnoses (T-cell lymphoma, skull base osteomyelitis, multiple myeloma-associated amyloidosis, granulomatosis with polyangiitis, urinary sepsis, neurosyphilis, and angle-closure glaucoma). False positives may be related to a condition responsible for an increase in intima-media thickness that produces a true halo sign but is not the consequence of GCA. Hypoechoic thickening of the arterial wall may indeed be related to cellular infiltration of another nature, or to parietal edema, as is the case in ANCA-associated vasculitis, skull base osteomyelitis, neurosyphilis, or lymphoma [20]. Hypoechoic thickening may also be due to the deposition of inert material in the arterial wall, as in amyloidosis [21,22,23]. These false positives may also be due to errors in image interpretation (i.e., absence of halo sign) because there is a learning curve in the performance and interpretation of TA ultrasound. Indeed, the TABUL study showed that the sensitivity of TA Doppler ultrasound for the diagnosis of GCA increased from 45% to 62% after 10 TA Doppler ultrasounds were performed [24].

Some GCA patients are characterized by an isolated involvement of large vessels without cranial involvement and thus correspond to the second phenotype of GCA that is entitled “large vessel vasculitis (LVV)-GCA” [11]. These patients are usually younger than cephalic GCA and have a less ischemic complication (vision loss or stroke) but a higher risk of relapse [11]. They are also characterized by an increased risk of aortic complication or any cardiovascular event [25,26,27]. The aorta is the artery most often affected in these patients. This does not usually cause any symptoms but can lead to complications in the long term, such as aneurysms and more rarely aortic dissection [16,28]. In patients with GCA, the risk of aortic aneurysm is indeed two times higher than in the general population [28]. Aortic aneurysms usually occur after several years of evolution, in a segment previously affected by the vasculitis, which is more frequently the thoracic aorta than the abdominal aorta, contrary to the general population [25]. After the aorta, the other most frequently affected vessels are: subclavian arteries (26–42.5%), carotid arteries (35–40%), femoral arteries (30–37%), and axillary arteries (14–17.5%), with frequencies varying according to the mode of recruitment and the imaging technique [17,29]. Patients with GCA have an increased risk of cardiovascular events: stroke, myocardial infarction, and lower limb arteritis [30]. Stroke is usually related to the involvement of the vertebrobasilar system by vasculitis [7,8,9]. Conversely, myocardial infarction is not related to coronary artery damage but to a mismatch between myocardial oxygen supply and demand that is probably related to systemic inflammation in these elderly patients [31].

There is no specific biological marker for GCA but patients almost always have elevated inflammatory proteins and increased sedimentation rates (ESR). Inflammatory anemia (hemoglobin <12 g/dL) is observed in 54.6% of cases and thrombocytosis in 48.8% of cases [32]. Nevertheless, ESR may be less than 50 mm/h in 10.8% of cases and its normality does not exclude the diagnosis [33], mainly in the case of ischemic signs [34]. 

The gold standard for the diagnosis of GCA is TA biopsy (TAB) that reveals a non-necrotizing granulomatous panarteritis, with an inflammatory cellular infiltrate composed of mononuclear cells (T lymphocytes and macrophages), sometimes giant cells, fragmentation of the internal elastic lamina (IEL), and destruction of the media and hyperplasia of the intima which induces the stenosis of the vascular lumen (Figure 1E–G) [35,36]. To confirm the diagnosis of GCA, the essential element is the presence of an inflammatory infiltrate within the media and/or intima. Elastophagy of the IEL and/or giant cells are pathognomonic but inconsistent. Intimal hyperplasia, which increases in frequency with age, and IEL dissociation are not specific to GCA. Isolated involvement of the adventitia and/or *vasa vasorum* is of uncertain diagnostic significance, as discussed below. The diagnosis of GCA may be retained even if the TAB does not show vasculitis lesions. Indeed, the sensitivity of TAB varies from 60% to 80% [2,37] and some patients have an extracephalic GCA that predominates on the aorta, subclavian, and/or carotid arteries so that, in this group of patients, only 52% of TAB are positive [38]. 

Glucocorticoids (GC) are the cornerstone of treatment for GCA. This treatment is remarkably effective but should be prescribed in high doses (40–80 mg/day prednisone-equivalent) at diagnosis for induction of remission and prevent ischemic complications [39,40,41]. In patients with GCA with acute visual loss or amaurosis fugax, the administration of 250 to 1000 mg intravenous methylprednisolone for up to 3 days should be considered [39,40,41]. Once remission is achieved, the dose of prednisone is then gradually reduced, first rapidly to a target dose of 15–20 mg/day within 2–3 months and then more slowly to target ≤5 mg/day after 1 year and to be stopped after two years [39,40,41]. 

When GC are tapered, about half of the patients will relapse, on average 7 months after diagnosis and at a mean dose of 7.5 mg/d of prednisone. These relapses are severe in only 3.3% of cases and it is exceptional to see ischemic complications on these occasions. The risk of relapse increases in the case of large-vessel involvement and in the case of a rapid decrease in GC therapy [42,43]. These relapses are easily controlled by increasing the dose of GC but contribute to exposing these patients to high cumulative GC doses and thus increase the risk of GC-induced adverse events, such as osteoporosis, fractures, diabetes, cardiovascular disease, or glaucoma [44]. It is commonly estimated that 86% of patients will have at least one side effect from steroids after one year of treatment [44]. It is therefore important to reduce the total dose and duration of GC without increasing the risk of relapse in GCA. Methotrexate and tocilizumab, an anti-interleukin-6 (IL-6) receptor monoclonal antibody, are the two drugs used in patients with relapses or steroid-related side effects [45,46]. The efficacy of tocilizumab to induce and maintain remission of GCA and to significantly spare GC has been shown in two recent randomized controlled clinical trials [46,47]; so that recent American recommendations propose to use a combination of prednisone and tocilizumab as first-line therapy in new-onset GCA [40]. In patients receiving GC-sparing therapy, faster GC taper and earlier withdrawal of GC should be considered on an individual basis. Data on the efficacy of methotrexate in GCA are more contradictive and only focused on newly diagnosed GCA patients [48,49,50]. However, a meta-analysis showed that methotrexate reduced the risk of relapse and spared glucocorticoids with an effect size that appears to be smaller than that of tocilizumab but needs to be reassessed in a comparative study [45]. This is what the Multicenter, Randomized, Controlled Trial (METOGiA) trial (NCT03892785) is currently investigating in France. Other treatments have been investigated in GCA: anti-tumor necrosis factor-alpha [TNF-α] blockers are not effective [51,52,53], neither is azathioprine [54]. Abatacept, a fusion protein of extracellular part of cytotoxic t-lymphocyte antigen-4 (CTLA-4) and Fc fragment of an IgG that blocks T-cell activation, has been reported to have a mild efficacy to maintain remission in a phase 2 trial [55] and is currently under investigation in a phase 3 trial (NCT04474847). Ustekinumab, an anti-p40 subunit monoclonal antibody that targets IL-12 and IL-23 pathways, has been reported to spare glucocorticoids in cohort studies [56,57], which was not confirmed in a pilot study with a very rapid glucocorticoid tapering regimen [58]. Ustekinumab is currently evaluated in a phase 2 study in France (NCT03711448). Mavrilimumab (anti-Granulocyte Macrophage Colony Stimulating Factor [GM-CSF] monoclonal antibody) and secukinumab (anti-IL-17 monoclonal antibody) have also been reported as promising therapies for GCA in phase 2 trials whose results have been presented at international conferences [59,60]. Secukinumab is now evaluated in an ongoing phase 3 trial (NCT04930094). Current clinical trials are also evaluating upadacitinib (Janus activated kinase-1 [JAK1] inhibitor, NCT03725202) and guselkumab (anti p19 subunit, NCT04633447). 

The pathophysiological mechanisms of GCA are becoming clearer but the triggering factor of this vasculitis has not been identified yet. It is likely that an agent, infectious or not, activates the Toll-like receptor of dendritic cells located in the adventitia and then leads to their activation and the recruitment of T cells and monocytes and finally to the formation of a granulomatous vasculitis with an intense vascular remodeling process [61]. Several infectious agents have been associated with the occurrence of GCA but none has been actually confirmed [62]. The most recent is the varicella-zoster virus, which is probably more consistent with another type of vasculopathy that we will discuss later in this review. A recent study has also reported on a defect in programmed cell Death protein ligand-1 (PD-L1) expression by vascular dendritic cells in GCA thus resulting in sustained activation of PD-1+ T cells and a loss of tolerance leading to vasculitis [63,64]. The involvement of this signaling pathway is highlighted by the description of a few cases of GCA occurring after treatment with immune checkpoint inhibitors (ICI) for cancer. The clinical presentation was mainly ophthalmologic (transient diplopia, amaurosis, or blindness). In most cases, TAB confirmed the diagnosis of GCA by showing typical granulomatous vasculitis lesions [65,66]. ICIs are probably a trigger rather than a differential diagnosis of GCA [64]. Treatment is the same as for GCA with the addition of discontinuation of the ICI. 

Like GCA, Takayasu arteritis (TAK) is a granulomatous large vessel vasculitis involving the aorta and its major branches. TAK can be distinguished from GCA by several epidemiological, clinical, arterial distribution, and therapeutic features. TAK is the second type of primitive large-vessel vasculitis. It is a much rarer disease than GCA, which mainly occurs in females (F/H sex ratio = 9) and in patients aged <40 years. TAK shares many features with GCA but usually does not affect TA. The differential diagnosis depends primarily on sex and age at onset, particularly in GCA patients without temporal arteritis [67,68,69].

### 2.2. Differential Diagnoses of GCA 

Main differential diagnoses of GCA are summarized in Table 1.

#### 2.2.1. Necrotizing Vasculitis 

Necrotizing vasculitides include anti-neutrophil cytoplasmic antibody (ANCA)-associated vasculitides and periarteritis nodosa (PAN). ANCA-associated vasculitides (granulomatosis with polyangiitis, microscopic polyangiitis, eosinophilic granulomatosis with polyangiitis) belong to the group of small-vessel vasculitides of the Chapel Hill classification [70]. In a study of 354 TA with inflammatory lesions, 77.5% had transmural involvement almost always related to GCA, 9% peri-adventitial small vessel vasculitis, 6.5% adventitial *vasa vasorum* vasculitis, and 7% inflammation limited to the adventitia. Of the 322 patients analyzed, 317 had a final diagnosis of GCA, 3 had ANCA-associated vasculitis with TAB showing lesions of peri-adventitial small vessel vasculitis, and only one had PAN [22]. An early study suggested that 1.4% of patients with suspected GCA actually had necrotizing systemic vasculitis and that 4.5% of TABs with inflammatory lesions were in patients with systemic necrotizing vasculitis [71]. More recently, the French Vasculitis Study Group described the characteristics of 50 patients with temporal involvement related to ANCA-associated vasculitis: 88% had cephalic symptoms (68% headache, 44% jaw claudication, 44% scalp tenderness, 16% abolition of the temporal pulse). For 66% of them, the examination revealed atypical symptoms for GCA, such as ear, nose, and throat signs, or renal, pulmonary, or neurological involvement (mononeuritis), which led to a search for ANCA [72]. Histological examination of the TA of these patients showed lesions of vasculitis of the small peri-adventitial vessels without the involvement of the media and intima. Panarteritis may sometimes be present but, unlike GCA, fibrinoid necrosis may be observed within the arterial wall (Figure 3A) and cellular infiltrates rich in T lymphocytes, macrophages and also eosinophils [72]. When this cephalic presentation delays the diagnosis of ANCA vasculitis, the risk of therapeutic failure is important. Thus, ANCA-associated vasculitis should be discussed when GCA symptoms are atypical and/or when associated with corticosteroid resistance and/or early relapse. ANCA testing and careful analysis of TAB are helpful for making the diagnosis [72].

This distinction between GCA and ANCA-associated vasculitis is very important since the treatment of necrotizing vasculitis frequently requires the association of glucocorticoids and rituximab or cyclophosphamide to achieve remission before introducing a maintenance therapy with methotrexate, azathioprine, or rituximab to prevent relapses that are frequent and can be severe in these diseases [76].

In PAN, which has become a rare disease, histological examination typically demonstrates a necrotizing vasculitis with transmural involvement of the temporal artery or one of its collaterals [22].

#### 2.2.2. Inflammatory Diseases Limited to the Adventitia 

The significance of inflammatory lesions limited to the adventitia is poorly understood. This type of involvement can be found during GCA but is not pathognomonic (Figure 3B,C). Restuccia et al. previously reported on the study of 455 TABs performed between 1983 and 2003, of which 16 showed periadventitial small-vessel vasculitis (SVV), 18 isolated vasa vasorum vasculitis (VVV), and 5 showed both. Compared with patients with classical GCA, these patients were therefore very rare and had less intense constitutional and cephalic symptoms together with a lower level of an acute-phase reactant at diagnosis. In addition, the initial and cumulative doses of prednisone were significantly lower in these patients compared to classical GCA, suggesting that SVV and VVV may be milder forms of GCA [77]. Another study including 80 patients with inflammation limited to adventitial and/or peri-adventitial tissue in TAB has shown that the diagnosis of GCA was significantly less frequently retained in these patients than in those whose TAB had shown transmural inflammation. Pathologic evidence of inflammation limited to adventitial and/or peri-adventitial tissue has therefore a low sensitivity and a low positive predictive value for the diagnosis of GCA [78]. Another study that analyzed 75 TA, including 31 with isolated adventitial involvement, showed that the positive predictive value for the diagnosis of GCA was only 17% in the case of vasculitis limited to the adventitia, 0% in the case of *vasa vasorum* vasculitis, and 7% in the case of peri-adventitial small vessel vasculitis. Moreover, this type of lesion does not seem to be associated with a particular clinical phenotype since the other 24 patients had a wide range of diagnoses such as idiopathic aortitis, seronegative arthritis, Sjögren’s syndrome, systemic lupus, Buerger’s disease, amyloidosis, neoplasia or various infections, and sometimes even no acute pathology [79]. The prognosis of these patients with isolated adventitial inflammation does not seem to be altered since Jia et al. showed that survival and the risk of ischemic complications after 5 years of follow-up were not increased in 56 patients with inflammation limited to the adventitia and who did not receive any corticosteroid therapy versus 39 control subjects [80].

#### 2.2.3. IgG4-Related Disease

IgG4-related disease (IgG4-RD) is a systemic inflammatory and fibrosing disease characterized by an infiltration of affected tissues by IgG4-producing plasma cells and increased serum IgG4 concentration (>1.35 g/L). The clinical manifestations of this disease are highly protean. Among the most frequent disorders are pancreatitis, fibro-inflammatory pseudotumors (pancreatic, orbital, salivary glands), sclerosing cholangitis, retroperitoneal fibrosis, or salivary gland damage (Mikulicz and Küttner syndromes). In addition, IgG4-RD can cause vasculitis that affects large and medium-sized vessels: abdominal aorta most frequently but also thoracic aorta and more rarely coronary, superior mesenteric, and iliac arteries [81]. Involvement of the TA is also possible and usually presents as an aneurysm of the TA with histological evidence of IgG4-RD. TA involvement may be isolated or associated with other vascular locations [82,83]. Histologically, there is parietal thickening predominantly in the adventitia with a lymphoplasmacytic infiltrate associated with fibrosis, numerous lymphoid follicles, and diffuse distribution of IgG4-positive plasma cells throughout the vessel wall.

#### 2.2.4. Sarcoidosis

Vasculitis is a rare manifestation of sarcoidosis with a predominance of large-vessel involvement, of which Takayasu or GCA are differential diagnoses to inflammatory aortitis and aneurysm formation [84,85]. Associations between sarcoidosis and GCA have also been described [85,86]. Sarcoidosis-associated aortitis can be difficult to distinguish from GCA when it occurs in a patient older than 50 years. Given the possible involvement of large vessels, pathological analysis of the TAB can reveal epithelioid and gigantocellular granuloma leading to the diagnosis of sarcoidosis (Figure 3D).

#### 2.2.5. VZV Vasculitis

Varicella-zoster virus (VZV) is a neurotropic virus whose primary infection is responsible for chickenpox and secondarily for zoster when viral replication resumes due to a diminished immune response. VZV is also capable of replicating in the arteries and inducing vasculitis [87]. This vasculitis, whose histological appearance is very similar to GCA, affects large and medium arteries and can lead to stenosis, occlusion, thrombosis, or dissection. In contrast to GCA, it can also affect small arteries, especially intracerebral arteries, and is, therefore, a rare cause of stroke (ischemic or hemorrhagic), intracerebral aneurysm, cerebral thrombophlebitis, spinal cord infarction, and cranial nerve damage [87,88]. Its diagnosis relies on clinico-radiological criteria, VZV PCR positivity, and/or intrathecal anti-VZV IgG synthesis, which is the virological test of choice for diagnosis. Clinically, VZV vasculitis is evoked by recent herpes zoster followed by the appearance of neurological symptoms, associated radiologically with the appearance of ischemic or hemorrhagic lesions in the brain, angiography showing “pearl necklace” cerebral arteries, and cerebrospinal fluid pleocytosis. The diagnosis of VZV vasculitis is not easy because the neurological disease often develops weeks and sometimes months after zoster, or even in the absence of any rash so that ischemic strokes are often attributed to atherosclerosis rather than viral infection [87]. VZV vasculitis can also reach the TA and mimic GCA by causing GCA-like symptoms. Recently, Gilden et al. suggested that VZV may be the triggering agent of GCA because, in contrast to older studies [62], they detected the presence of VZV antigens in 73% of positive and 64% of negative TAB from patients with GCA and only 22% of negative TAB from controls [89]. These authors also reported observations of patients with GCA symptoms that showed little or no improvement on corticosteroid therapy and resolved after treatment with acyclovir [89]. However, reports of successful acyclovir treatment do not include the typical GCA conditions (negative or atypical TAB, coexisting necrotizing VZV retinitis, ophthalmic herpes zoster, or meningitis with anti-VZV IgG in the CSF) [89]. Thus, VZV vasculitis should rather be considered as a differential diagnosis of GCA.

## 3. Others Temporal Artery Vascular Diseases

### 3.1. Post-Traumatic Complications of the Temporal Arteries

Trauma to the periauricular region can cause a false aneurysm or arteriovenous fistula of the TA [90,91,92]. A false aneurysm is a rare event occurring after a localized rupture of the vessel wall and consists of a blood bag communicating with the injured vessel and contained by the adjacent tissues. Doppler ultrasonography reveals an anechoic, circulating formation in contact with the TA and whose pertus is derived from the latter [92]. The arteriovenous fistula of the scalp can also be of congenital origin. These lesions of the TA cause the appearance of a pulsatile mass, associated with headaches or pulsatile tinnitus [90]. Treatment options include surgical excision, ligation of the feeding vessels, or the use of endovascular techniques (percutaneous embolization) [90,91,92].

The TA may be the site of arteriovenous malformations, which are rare, usually congenital vascular anomalies, composed of a complex network of interconnected arteries and veins. Up to 20% of arteriovenous malformations involving the scalp are associated with a history of trauma [93]. Arteriovenous malformations of the head and neck occur in 0.1% of the population and TA is involved in 75% of scalp arteriovenous malformations [94]. Clinically, the patient may present with headache, local pain, paresthesias, tinnitus, ischemic necrosis, or nidus ulceration and hemorrhage associated with *thrill* on palpation [94].

### 3.2. Atheromatous Disease

The existence of atheromatous lesions is almost constant at the age when GCA is revealed [2]. The Atheromatous disease leads to the appearance of more or less calcified atheromatous plaques in the arterial wall, which on ultrasonography are manifested by a thickening of the intima-media thickness, generally hyperechoic or isoechoic, more rarely hypoechoic. This appearance may be confused with a true halo sign (Figure 2). De Miguel et al. showed in a cohort of 40 patients over 50 years of age at high cardiovascular risk and without GCA, that an intima-media thickness at the level of the common carotid artery > 0.9 mm on Doppler ultrasound was associated with an increase in intima-media thickness at the level of the TA > 0.3 mm and could therefore be confused with a halo sign [95]. As a result, they proposed to retain the threshold of 0.34 mm of intima-media thickness in at least two branches of the TA (common, frontal, or parietal) to define a true halo sign. This threshold made it possible to exclude 97.5% of patients with atheromatous damage to the TA [95].

### 3.3. Calcifying Uremic Arteriolopathy (Calciphylaxis)

Calcifying uremic arteriolopathy (calciphylaxis) mainly affects patients with chronic end-stage renal disease and leads to the development of calcifications in arterioles and soft tissues. Arteriolar calcifications are the cause of an obliterative vasculopathy leading to ischemia of the perfused territory (mainly dermis and hypodermis) and then to its necrosis. Clinically, the skin lesions are very painful, evolve towards ulceration, and are at high risk of infection. Histologically, calciphylaxis is a vasculopathy without vasculitis, characterized by calcification predominantly in the media with narrowing of the vascular lumen and the presence of fibrin microthrombi [96]. A few cases of calciphylaxis of TA have been described, mimicking GCA with ophthalmologic involvement (decreased visual acuity, sudden blindness, diplopia). An ophthalmologic examination may demonstrate authentic anterior ischemic optical neuropathy. Although GCA is more common in women, the majority of cases of TA calciphylaxis occur in men around 70 years of age, who are hypertensive and have renal insufficiency [97].

## 4. Conclusions

The term temporal arteritis should no longer be used to refer to GCA because, although cephalic GCA remains the main cause of TA vasculitis, other vasculitides can affect this artery such as ANCA-associated vasculitis, PAN, or VZV-induced vasculitis. The TA can also be the site of noninflammatory temporal vasculopathies such as atheroma, calciphylaxis, or posttraumatic complications. A careful clinical examination, the search for ANCA, the performance of the echo-Doppler by a trained operator, and the performance of a TAB help clinicians make the correct diagnosis.

## Figures and Tables

**Figure 1 jcm-11-00275-f001:**
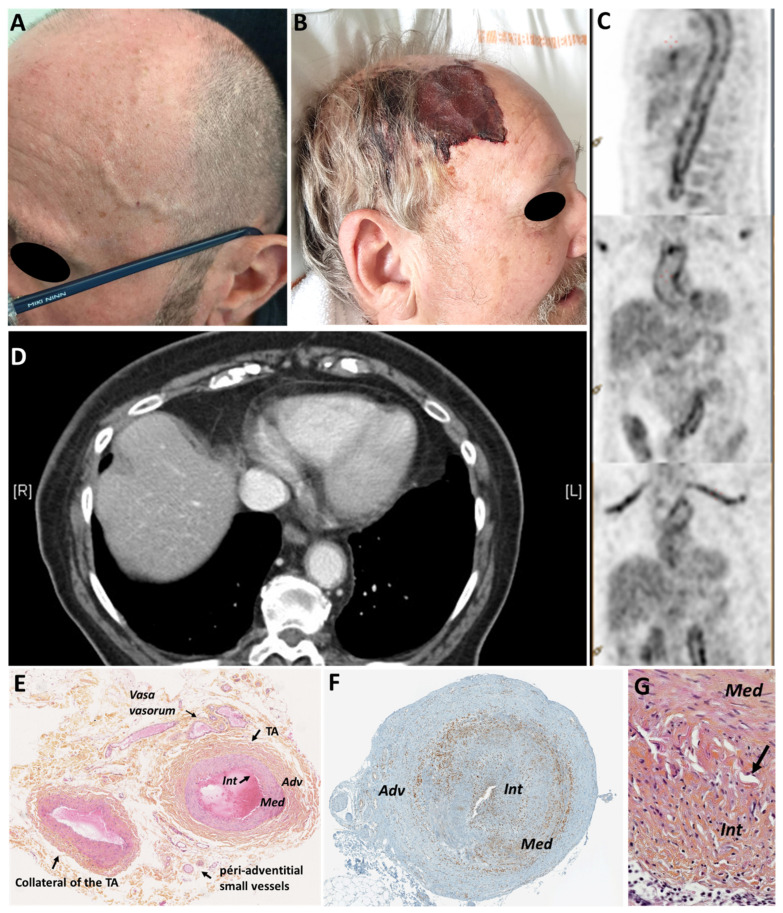
(**A**). Abnormal left temporal artery (enlarged, tender, and without pulse) in a giant cell arteritis (GCA) patient. (**B**): Scalp necrosis in a patient with severe cephalic GCA (**C**): PET-CT showing large vessel involvement (aorta and subclavian arteries) in a GCA patient. (**D**): Angio-CT showing aortitis in a GCA patient. The wall of the aorta is thickened (>2 mm) in a circumferential and homogeneous manner (white arrow). (**E**): Histological sections of a healthy temporal artery biopsy (TAB). The structure of the media is preserved, the intima is thin, the internal elastic lamina is preserved, the wall is not infiltrated by mononuclear cells and the vascular lumen is preserved. *Vasa vasorum* are seen in the adventitia and small peri-adventitial vessels are observed around the adventitia. A collateral of the TA is also observed; (**F**): Immunohistochemistry showing T-cell labeling (CD3) in a positive TAB of a GCA patient. Marron staining shows typical transmural T-cell infiltrate with a predominance at the adventitia/media junction. The media and the internal elastic lamina are destroyed, the intima is hyperplastic and the vascular lumen is stenosed. (**G**): Fragmentation of the internal elastic lamina (arrow) in a GCA-positive TAB. Adv: adventitia; TA: temporal artery; TAB: temporal artery biopsy; Int: intima; Med: media.

**Figure 2 jcm-11-00275-f002:**
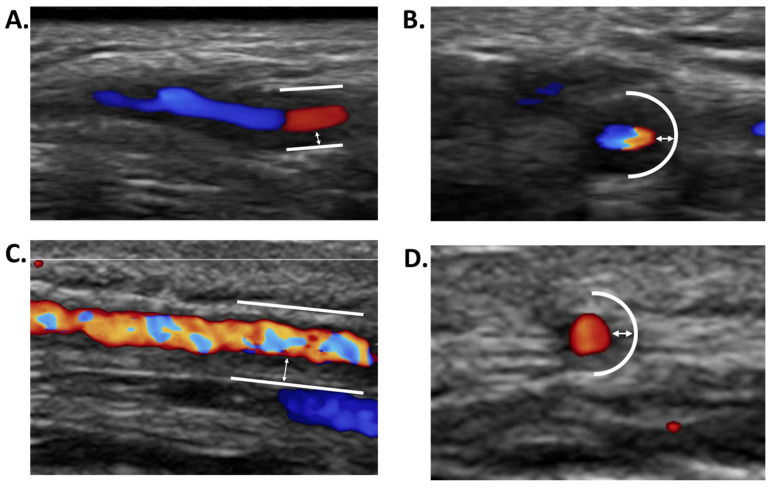
(**A**,**B**): True halo sign: hypoechoic parietal thickening of the temporal artery wall on Doppler ultrasound in a patient with GCA, in longitudinal (**A**) and transverse (**B**) sections; (**C**,**D**): iso/hyperechogenic parietal thickening of the temporal artery wall related to atherosclerosis, in longitudinal (**C**) and transverse (**D**) sections. The solid line shows the adventitia (hyperechoic), the double arrow shows the intima-media thickness, the arterial lumen is colored.

**Figure 3 jcm-11-00275-f003:**
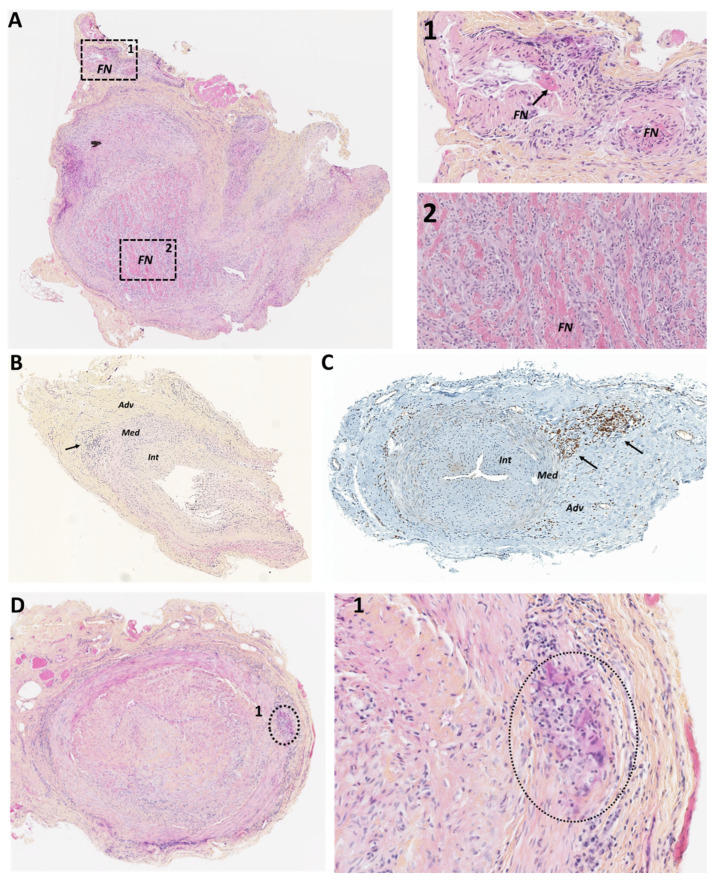
Histological sections of temporal artery biopsies (TAB). (**A**): TAB showing fibrinoid necrosis within the temporal artery (TA) (A1) and in the peri-adventitial *vasa-vasorum* (A2); (**B**): TAB showing vasculitis limited to the adventitia (arrow); (**C**): immunohistochemistry showing T-cell (CD3) labeling in a TAB with vasculitis limited to the adventitia (arrow). (**D**): TAB from a patient with sarcoidosis showing an epithelioid and gigantocellular granuloma at the media-adventitia junction. Adv: adventitia; TA: temporal artery; Int: intima; Med: media; NF: fibrinoid necrosis.

**Table 1 jcm-11-00275-t001:** Temporal arteritis.

	Epidemiology	Cephalic Clinical Signs	Extra-Cephalic Clinical Signs	Biology	Histological Signs (TAB)	Treatment
**GCA**	Incidence: 14.6 (range: 6.0–43.6) per 100,000 persons aged ≥50 years [73] Prevalence: 75.5 per 100,000 inhabitants (107.8 in women, 40.1 in men) [74]	- Headache, scalp tenderness, or jaw claudication.Abnormal TA exam: induration, tenderness to palpation, edema, decrease/abolition of the temporal pulses - No involvement of the intracerebral vessels.	- PMR - Claudication of a limb - Abolition/decrease of a peripheral pulse - Aortic complication (aneurysm, dissection) - Ischemic complications: Ophthalmological: AAION, CRAO, PION, diplopia Neurological: Stroke (posterior vertebral territory) Myocardial infarction, limb ischemia	- Inflammatory syndrome (95%) - Inflammatory anemia, thrombocytosis (40–50%)	Granulomatous, non-necrotizing panarteritis, inflammatory cellular infiltrate of the media and/or intima made of mononuclear cells, multinuclear giant cells, fragmentation of the internal elastic lamina, destruction of the media, stenosing hyperplasia of the intima	- Glucocorticoids (constant effectiveness except for ischemic sequelae) - 2nd line (relapse(s), corticodependence): tocilizumab, methotrexate
**Necrotizing vasculitis (AAV and PAN)**	**PAN**: annual incidence = 0.9–8.0 per million in Europe prevalence = 31 per million [73] **AAV**: Combined incidence annual rate for GPA, MPA and EGPA = 24.7 to 33.0 per million [73]	- Cephalic symptoms in 88% of cases in case of temporal involvement (headache, jaw claudication, scalp tenderness, abolition of the temporal pulse). - Other systemic signs (renal, skin, peripheral neuropathy, ENT...)	- Arthralgia, myalgia - Skin signs (purpura, necrosis, livedo) - ENT signs (rhinitis, sinusitis) - Ophthalmologic signs (vasculitis, exophthalmia, and scleritis in GPA) - peripheral neuropathy (mononeuritis > polyneuritis) - Glomerular nephropathy (AAV) or vascular nephropathy (PAN) - Pachymeningitis (GPA) - Pulmonary signs (asthma (EGPA), nodules (GPA), alveolar hemorrhage (GPA and MPA)) - Heart disease (EGPA) - Digestive signs (perforation, pancreatitis, appendicitis, peritonitis), mainly in PAN	- Inflammatory syndrome - Biological signs related to organ damage - ANCA *Anti-PR3: GPA* *Anti-MPO: EGPA, MPA* *No ANCA: PAN*	Necrotizing vasculitis (fibrinoid necrosis) with cellular infiltrate (T lymphocytes, macrophages, neutrophils, and/or eosinophils): - AAV: small vessels (vasa vasorum or peri-adventitial vessels) without the involvement of the media and intima - PAN: temporal artery and/or its collaterals	- Corticosteroids - Immunosuppressants (rituximab, cyclophosphamide, methotrexate)
**IgG4-RD**	Prevalence = 6/100,000 inhabitants [75]	Aneurysm of the TA, isolated or associated with other vascular localizations (large vessel vasculitis, aortitis).	- retroperitoneal fibrosis - adenopathies - pseudo-tumors - pancreatitis - cholangitis - dacryoadenitis, orbital pseudotumor - involvement of the salivary glands (Mikulicz, Küttner) - pachymeningitis - interstitial lung disease - glomerular or tubulointerstitial nephropathy	- Biological signs related to organ damage - Mild inflammatory syndrome (30–40%) - Hypereosinophilia (30–40%) - Hypocomplementemia (30%) - Polyclonal hypergammaglobulinemia (70%) - Elevation of serum IgG4 (>80%) - Elevation of total IgE (>80%)	Parietal thickening of the adventitia, lymphoplasmacytic and eosinophilic infiltrate, +/- storiform fibrosis, numerous lymphoid follicles, IgG4-positive plasma cells disseminated throughout the vessel wall, arterial thrombosis	- Corticosteroid therapy (constant effectiveness) - 2nd line: rituximab - 3rd line: azathioprine, mycophenolate mofetil
**VZV**	- No specific data, very rare	- Headaches - Possible involvement of intracerebral arteries (ischemic or hemorrhagic stroke, intracerebral aneurysm, cerebral thrombophlebitis, spinal cord infarction, cranial nerve involvement)	- Zoster +/− recent or ophthalmic shingles - VZV necrotizing retinitis - Ischemic optic neuropathy - Other arterial attacks of VZV vasculitis (stenoses, occlusions, thromboses, dissections of large and medium caliber arteries)	- Positive VZV PCR (CSF) - Measurement of anti-VZV IgG and IgM in serum and CSF (intrathecal synthesis of anti-VZV IgG)	Granulomatous arteritis, often transmural inflammation, media necrosis, multinucleated giant cells, epithelioid macrophages combined with the presence of VZV antigens	- Acyclovir 10–15 mg/Kg/8 h IV for 14 days - No effectiveness of corticosteroids

AAION: acute anterior ischemic optic neuropathy; AAV: ANCA-associated vasculitis; ANCA: neutrophil cytoplasmic antibody; CRAO: central retinal artery occlusion; CSF: cerebrospinal fluid; EGPA: eosinophilic granulomatosis with polyangiitis (Churg-Strauss disease); GCA: giant cell arteritis; GPA: granulomatosis with polyangiitis (Wegener’s disease); IgG4-RD: IgG4-related disease; MPA: microscopic polyangiitis; PION: posterior ischemic optic neuropathy; PMR: polymyalgia rheumatica; TAB: temporal artery biopsy; VZV: varicella-zoster virus. PAN: periarteritis nodosa; ENT: ear nose and throat; MPO: myeloperoxidase.

## Data Availability

Not adapted.

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
