# Peer review of "Temporal Artery Vascular Diseases"

_jcm, 2022, doi:10.3390/jcm11010275_

Round 1
Reviewer 1 Report
Dear Authors,
I read with interest your review article, where you discussed the various temporal artery diseases.
Here are my comments and suggestions.
1) The text should be structured in a different way. In particular, paragraph 3.4 (A halo sign without GCA) should be placed in page 3, before "Some GCA patients are characterized by an isolated.....". References should be changed accordingly;
2) Please, consider adding this reference: Manzo C, Reumatologia 2016; doi: 10.5114/reum.2016/63.663;
3) Data regarding the efficacy of biological DMARDS different from tocilizumab in patients with GCA are lacking. Please, discuss these literature data;
4) Paragraph no. 2.2.5 (Iatrogenic temporal arteritis) may be misleading, because the relationship between GCA and ICIs therapy is still under discussion. This paragraph should be re-written in a more extensive way, and with more specific references ;
5) Conclusions: the term temporal arteritis cannot be referred to non-inflammatory temporal vasculopathies. Please, modify this sentence.
Author Response
Reviewer #1:
1) The text should be structured in a different way. In particular, paragraph 3.4 (A halo sign without GCA) should be placed in page 3, before "Some GCA patients are characterized by an isolated.....". References should be changed accordingly;
We thank reviewer 1 for this comment and modified the manuscript as requested.
2) Please, consider adding this reference: Manzo C, Reumatologia 2016; doi: 10.5114/reum.2016/63.663;
We thank reviewer 1 for this comment. This reference was added in paragraph 1.1.
3) Data regarding the efficacy of biological DMARDS different from tocilizumab in patients with GCA are lacking. Please, discuss these literature data.
We agree with reviewer 1 that these data were lacking. We added available data about DMARDS and biologics that have been evaluated in GCA and also detailed ongoing therapeutic trials.
4) Paragraph no. 2.2.5 (Iatrogenic temporal arteritis) may be misleading, because the relationship between GCA and ICIs therapy is still under discussion. This paragraph should be re-written in a more extensive way, and with more specific references.
We thank reviewer 1 for this comment with which we agree. To avoid confusion, this section has been integrated into the paragraph on GCA with a focus on the involvement of the PD-1/PD-L1 pathway in the pathophysiology of GCA.
5) Conclusions: the term temporal arteritis cannot be referred to non-inflammatory temporal vasculopathies. Please, modify this sentence.
We agree with reviewer 1 that this concluding sentence is confusing, as requested, it was rephrased for the following: ““The term temporal arteritis, should no longer be used to refer only to GCA because, although cephalic GCA remains the main cause of TA vasculitis, other vasculitis may affect this artery such as ANCA-associated vasculitis, PAN or VZV-induced vasculitis. The TA may also be the site of non-inflammatory temporal vasculopathies such as atheroma, calciphylaxis or post-traumatic complications. A careful clinical examination, the search for ANCA, the performance of the echo-Doppler by a trained operator, and the performance of a TAB help clinicians make the correct diagnosis.”
Reviewer 2 Report
The manuscript entitled Temporal Artery Vascular Diseases is a comprehensive, well written review about the differential diagnosis of Giant Cell Arteritis.
The part about GCA is concise, rich in information. There are two typos on page 3, line 16 pair instead of nerve and last line of third para TEP instead of PET.
Regarding figure 1, I suggest to focus on the findings (Arteria in A, Necrosis in B and Aortitis in D) and to enlarge these. In C there is no need to show the middle and right images. It would be more informative to get one or two classical findings.
Table 1 is difficult to read. I suggest to print this on two pages and to draw vertical lines for better readability.
Suggest to add the prevalence of the differential diagnoses
Treatment of GCA: I am aware that many French experts prefer MTX to Tocilizumab. However, based on the current knowledge, I can not agree that MTX/TCZ are proposed in case of relapse. In case of risks of GC side effects (osteoporosis, Diabetes mellitus, arteriosclerosis) most experts agree on early prescription of Gc-sparing drugs (see also EULAR recommendations and recent results of the GUSTO study (Lancet Rheumatology), which shows an additional GC-sparing effect of TCZ; (maybe add ref.))
Necrotizing vasculitis: It is confusing to describe ANCA associated and PAN in the same paragraph. I strongly recommend to split it.
Regarding clinical signs of necrotizing vasculitis: Add Mononeuritis, it is more prevalent than polyneuritis
IgG4 related diseases: add MMF as treatment option with references.
2.2.2. suggest to add and discuss references of Salvarani. He described the different histological patterns of GCA.
Conclusions: I agree that the term temporal arteritis should not be used as a diagnostic label. Nevertheless it remains correct if it describes an inflammation of the temporal artery. Suggest to propose replacing the diagnostic term by cGCA for cranial GCA.
Author Response
Reviewer #2:
1) Regarding figure 1, I suggest to focus on the findings (Arteria in A, Necrosis in B and Aortitis in D) and to enlarge these. In C there is no need to show the middle and right images. It would be more informative to get one or two classical findings.
We have taken into consideration the comments of reviewer 2 and modified the figure accordingly.
2) Table 1 is difficult to read. I suggest to print this on two pages and to draw vertical lines for better readability.
We have taken this comment into account and tried to make table 1 more readable.
3) Suggest to add the prevalence of the differential diagnoses
We thank reviewer 2 for this comment. Available data were added in Table 1.
4) Treatment of GCA: I am aware that many French experts prefer MTX to Tocilizumab. However, based on the current knowledge, I can not agree that MTX/TCZ are proposed in case of relapse. In case of risks of GC side effects (osteoporosis, Diabetes mellitus, arteriosclerosis) most experts agree on early prescription of Gc-sparing drugs (see also EULAR recommendations and recent results of the GUSTO study (Lancet Rheumatology), which shows an additional GC-sparing effect of TCZ; (maybe add ref.))
We agree with reviewer 2 regarding this comment. Details regarding currently available or developing sparing therapies in GCA have been added in the manuscript. In addition, we made it clear that the level of evidence was lower for MTX than for TCZ.
5) Necrotizing vasculitis: It is confusing to describe ANCA associated and PAN in the same paragraph. I strongly recommend to split it.
We agree with reviewer 2 that PAN is different from AAV. The studies we report in the paragraph about necrotizing vasculitis have mixed cases of PAN and AAV so that it is difficult to separate this paragraph into two parts without repetition. We have therefore modified the organization of the paragraph so that the elements on the PAN appear at the end and so that the part on the AAV is more prominent.
6) Regarding clinical signs of necrotizing vasculitis: Add Mononeuritis, it is more prevalent than polyneuritis
We totally agree with this comment. It was added in table 1
7) IgG4 related diseases: add MMF as treatment option with references.
It was added in table 1.
8) 2.2.2. suggest to add and discuss references of Salvarani. He described the different histological patterns of GCA.
Two references of the team of C Salvarani were already cited in the first version of the manuscript:
- Ref 77: Galli, E.; Muratore, F.; Boiardi, L.; Restuccia, G.; Cavazza, A.; Catanoso, M.; Macchioni, P.; Spaggiari, L.; Casali, M.; Pipitone, N.; et al. Significance of inflammation restricted to adventitial/periadventitial tissue on temporal artery biopsy. Semin Arthritis Rheum 2020, 50, 1064-1072, doi:10.1016/j.semarthrit.2020.05.021.
- Ref 22: Cavazza, A.; Muratore, F.; Boiardi, L.; Restuccia, G.; Pipitone, N.; Pazzola, G.; Tagliavini, E.; Ragazzi, M.; Rossi, G.; Salvarani, C. Inflamed temporal artery: histologic findings in 354 biopsies, with clinical correlations. Am J Surg Pathol 2014 Oct, 38, 1360-1370.
As requested by reviewer 2, we also added this reference from Salvarani’s team: Restuccia, G.; Cavazza, A.; Boiardi, L.; Pipitone, N.; Macchioni, P.; Bajocchi, G.; Catanoso, M.G.; Muratore, F.; Ghinoi, A.; Magnani, L.; et al. Small-vessel vasculitis surrounding an uninflamed temporal artery and isolated vasa vasorum vasculitis of the temporal artery: two subsets of giant cell arteritis. Arthritis and rheumatism 2012, 64, 549-556, doi:10.1002/art.33362.
This reference is discussed in the dedicated paragraph.
9) Conclusions: I agree that the term temporal arteritis should not be used as a diagnostic label. Nevertheless, it remains correct if it describes an inflammation of the temporal artery. Suggest to propose replacing the diagnostic term by cGCA for cranial GCA.
We fully agree with the reviewer 2 and have changed the conclusion to make it clearer
“The term temporal arteritis, should no longer be used to refer to GCA because, although cephalic GCA remains the main cause of TA vasculitis, other vasculitis may affect this artery such as ANCA-associated vasculitis, PAN or VZV-induced vasculitis. The TA may also be the site of non-inflammatory temporal vasculopathies such as atheroma, calciphylaxis or post-traumatic complications. A careful clinical examination, the search for ANCA, the performance of the echo-Doppler by a trained operator, and the performance of a TAB help clinicians make the correct diagnosis.”
Round 2
Reviewer 1 Report
Dear Authors,
all my suggestions were satisfactorily met in the revised version of your article.
Reviewer 2 Report
it is easier for a reviewer to judge the changes if they are displayed in the answers or at least readily identifiable by mentioning the paragraph and the lines
I guess the second reviewer asked for adding the preliminary data about drugs currently being tested. however, for a clinician it would be more helpful to learn about additional data of licensed drugs.
I have no new comments